# A Graphical Tool to Describe the Operating Point of Direct Reduction Shaft Processes

Thibault Quatravaux [†]

Institut Jean Lamour, UMR 7198 Université de Lorraine/CNRS, Campus Artem, 2 allée André Guinier-BP 50840, LabEx DAMAS, CEDEX 01, F-54011 Nancy, France; thibault.quatravaux@univ-lorraine.fr
[†] Formerly ArcelorMittal R&D Maizières Process.

**Abstract:** This article presents a new graphical tool for direct reduction shaft processes inspired by the Rist diagram developed for blast furnaces. The tool represents gas flows using vectors, with specific consumption and specific oxidation as components to indicate gas/iron ratios. Key features include consideration of gas chemical composition for vector directions, easy visual representation of gas mixtures, as well as reduction and carburization rates of direct reduced iron (DRI). The tool also includes thermodynamic conditions for reduction from the Chaudron diagram, analogous to the Rist diagram. Several practical applications are presented, including quantifying gas moisture, evaluating the measurement consistency of flowmeters and gas analyzers in top gas recycling, and evaluating instantaneous DRI production by analyzing reducing gas at the inlet and outlet of the shaft. This graphical tool could be useful for production teams to monitor and optimize process flows and promote understanding among students, engineers, technicians, and operators. Its potential for online use further enhances its practical value. As a result, the tool is of significant academic and industrial interest in improving process efficiency and optimization.

**Keywords:** direct reduction; ironmaking; Midrex; Rist diagram





## 1. Introduction

The production of iron by direct reduction is the main alternative to traditional blast furnace iron production. Over the past two decades, this process has gained considerable popularity. According to the World Steel Association [1,2], direct reduced iron (DRI) now accounts for 10% of total iron production, a significant increase from 1.5% in the 1980s, and reached 125 Mt in 2022, about three times the annual production in the early 2000s. Many current research efforts, both industrial and academic, are devoted to the study of the kinetics of direct reduction processes in which the reducing agent is wholly or partially composed of hydrogen [3–6]. This path is emerging as one of the most promising methods considered in the strategy for decarbonization of the steel industry [7,8].

Among the direct reduction processes, the gas-based shaft furnace is the most widely used technology, largely represented by the Midrex NG[TM] and Tenova HYL processes, which use natural gas as the reducing agent for iron oxides, as explained by Ghosh and Chatterjee [9] (2008).

As direct reduction is a relatively new family of processes in ironmaking, with the first reactors being less than 50 years old, there is little literature devoted to a detailed understanding of its operating point. It is interesting to look at the recent evolution of blast furnace operating points over the last few decades, which is the result of a series of complementary research studies carried out between the 1950s and the 1970s, which led to a paradigm shift in the approach to this process [10].

Among these notable studies, Rist and Bonnivard [11] (1963) introduced a graphical representation of the operating point of a blast furnace, incorporating a model of staggered iron oxide reduction mechanisms in a theoretical counter-current gas–solid reactor setup.

Rist and Bonnivard [12] (1966) further refined their approach by considering the specific operating conditions of the blast furnace. Several studies are based on the application of the diagram to the optimization of the operating point, such as the effect of wind injection conditions at the tuyeres by Rist and Meysson [13] (1965), or the determination of the minimum coke yield by the same authors [14] (1964), later also studied by Nicolle et al. [15] (1980).

This graphical tool provided visual support for the blast furnace modeling developed by Michard [16] (1961), which was based on the theoretical work provided by Kitaev in the 1950s. After 30 years of industrial and academic applications, Rist [17] (1992) concluded that this approach demonstrated the value of a synthetic description of the entire reactor system. The pedagogical benefits of this methodology were obvious, facilitating process understanding and the determination of likely operating points through an intuitive visual approach requiring minimum calculations. For these reasons, it remains a topical tool for the study and optimization of blast furnace operation [18–20], as well as for the evolution of the process towards decarbonization [21], by hydrogen injection or the use of biomass [22].

Quatravaux et al. [23] (2021) have recently successfully adapted the Rist operating diagram to direct reduction shaft processes, specifically for the Midrex NG$^{TM}$ process. They have shown that this type of approach remains relevant among the many tools currently available for process study, complementing the various numerical solutions, whether via 1D coupled reactor models provided by Parisi and Laborde [24] (2004) or Shams and Moazeni [25] (2015), as well as simulations of coupled phenomena computed by the finite volume method developed by Hamadeh et al. [26] (2018).

However, this diagram is only a partial description of the DRI process, as it deals only with the reduction zone of the shaft furnace. It hides part of the process including top gas recycling and reducing gas preparation.

Therefore, this manuscript presents a novel graphical tool developed specifically for direct reduction shaft processes which is applied to the description of the Midrex NG$^{TM}$ process. It provides a global approach to an industrial plant and incorporates many of the concepts found in the Rist operating diagram.

## 2. Modeling

### 2.1. Description of the Midrex NG$^{TM}$ Process

The main differences between direct reduction and blast furnace iron ore reduction are the use of natural gas as a reducing agent and as a heat source with an operating point at temperatures below 1000 °C, without melting the iron-bearing material. The Midrex NG$^{TM}$ process is the most widely used technology in the steel industry for the production of direct reduced iron (DRI), accounting for 60% of total DRI production. The Midrex NG$^{TM}$ process industrial unit is based on the combination of three main components: a shaft furnace, a reformer, and a heat recovery system. This configuration allows for optimum recovery of natural gas by fully recycling the gases from the shaft furnace into the process. Figure 1 shows the operating principle of the Midrex NG$^{TM}$ process.

The description of the operating point of this process is addressed through the main steps identified and described below:

- The top gas (**T**) exhausted from the shaft furnace, rich in CO and $H_2$, is completely recycled. A total of 30% of the top gas is recycled as a fuel gas (**F**) in the reformer and the heat recovery unit. The other fraction, called process gas (**P**), is recycled as a reducing agent.
- The fuel gas (**F**) is burned in the reformer burners with an additional amount of natural gas and preheated air from the heat recovery unit. The energy from the hot reformer exhaust gases is recovered in a special device.
- The reducing gas is prepared in several steps. First, the process gas (**P**) is mixed with injected natural gas. This mixture, called the feed gas, is preheated in the heat recovery unit and then injected into the reformer. Cracking takes place in the reformer tubes between $CH_4$, $CO_2$, and $H_2O$. The resulting reformed gas (**R**) is composed mainly of CO, $CO_2$, $H_2$, and $H_2O$, with a small amount of $CH_4$ remaining (few %).



- The reformed gas (**R**) is mixed with additional natural gas, called enrichment gas, and pure oxygen, to form the bustle gas (**B**), whose temperature can reach 900–950 °C.
- The bustle gas (**B**) is injected into the transition zone located at the mid-height of the shaft. Additional natural gas is also injected into the shaft, at the bottom of the cooling gas loop, and into the transition zone with the bustle gas.
- The shaft furnace is a vertical counter-current gas–solid reactor with a downward flow of iron oxides and an upward flow of hot reducing gas. The iron pellets are then both reduced and carburized. The direct reduced iron achieves a high degree of metallization (92–96%) and a moderate degree of carburization (2–2.5% of carbon in total mass).

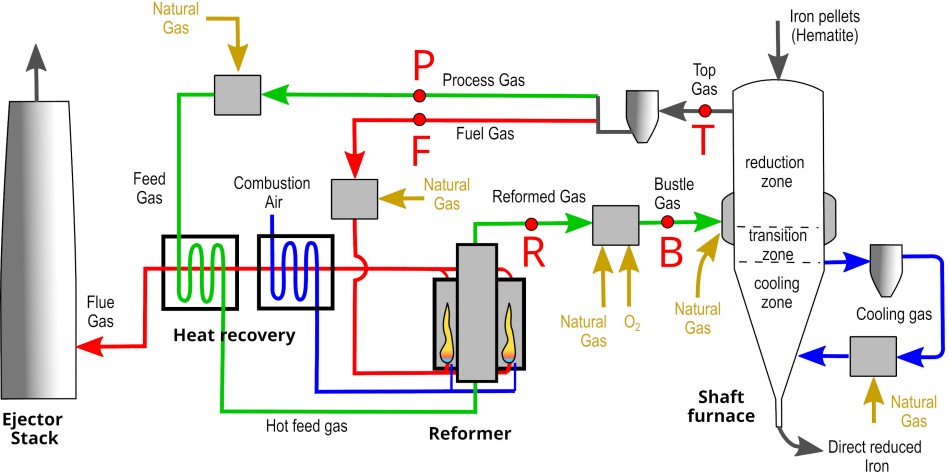

**Figure 1.** Operating principle of the Midrex NG$^{TM}$ process.

### 2.2. Definition of the Mass Balance Diagram

We will consider the characteristics of a gas flow (**G**) at each point of the installation. We will describe this gas in terms of adimensional variables that quantify some stoichiometric relationships between this gas flow and that of the iron-bearing material passing through the shaft furnace.

The first variable $\mu$, called specific consumption by Rist, represents the stoichiometric ratio between the carbon and hydrogen gas flow and the iron flow passing through the shaft furnace, defined by (1):

$$\mu = \frac{\phi_C^{gas} + \frac{\phi_H^{gas}}{2}}{\phi_{Fe}^s} \tag{1}$$

where $\phi_C^{gas}$ and $\phi_H^{gas}$ are the molar flux of carbon and hydrogen calculated with the following relation:

$$\phi_i^{gas} = \frac{Q_v^{gas}}{V_m} \times \sum_{molecule\ j} n_i^j \cdot \%j \tag{2}$$

$Q_v^{gas}$ is the volumetric flow rate of the gas, $V_m$ is the molar volume of the gas, $\%_j$ is the volumetric fraction of molecule $j$ in the gas, and $n_i^j$ is the number of atoms $i$ in the molecule $j$.

The gas is mainly composed of CO, $CO_2$, $H_2$, $H_2O$, and hydrocarbons $C_mH_n$; the molar gas fluxes can be approximated by the following relations:

$$\phi_C^{gas} = \frac{Q_v^{gas}}{V_m} \times \left( \%CO + \%CO_2 + \sum m \cdot \%C_mH_n \right) \tag{3}$$

$$\phi_H^{gas} = \frac{Q_v^{gas}}{V_m} \times \left( 2\%H_2 + 2\%H_2O + \sum n \cdot \%C_mH_n \right) \tag{4}$$

$$\phi_O^{gas} = \frac{Q_v^{gas}}{V_m} \times (\%CO + 2\,\%CO_2 + \%H_2O) \tag{5}$$

$\phi_{Fe}^s$ is the molar flux of iron from solid s into the furnace shaft. It is obtained by Equation (6) as a function of the mass flow rate $Q_m^s$ and the mass fraction of iron $w_{Fe}^s$ in the solid. The reference solid used in the calculation can be pellets or DRI.

$$\phi_{Fe}^s = Q_m^s \times \frac{w_{Fe}^s}{M_{Fe}} \tag{6}$$

Similarly, the specific oxidation $\nu$ is defined as the stoichiometric ratio of carbon and oxygen gas flux to iron flux through the shaft:

$$\nu = \frac{\phi_O^{gas} - \phi_C^{gas}}{\phi_{Fe}^s} \tag{7}$$

Rist defined the oxidation degree of a gas as the following stoichiometric ratio:

$$x = \frac{\phi_O^{gas} - \phi_C^{gas}}{\phi_C^{gas} + \frac{\phi_H^{gas}}{2}} \tag{8}$$

From this definition, we can deduce:

$$x = \frac{\nu}{\mu} \tag{9}$$

Finally, the two variables $\mu$ and $\nu$ are plotted on a diagram on the *x*-axis and *y*-axis, respectively. Any gas **G** flowing through the direct reduction unit can be plotted on such a graph, as shown in Figure 2.

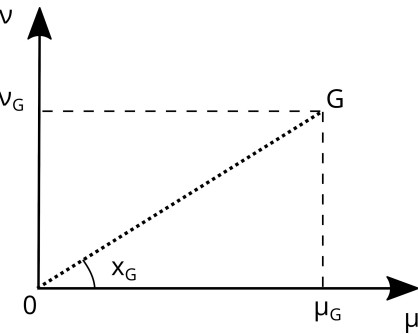

**Figure 2.** Graphical representation of a gas **G** in the mass balance diagram.

### 2.3. Properties of the Mass Balance Diagram

From the definitions given in Section 2.2, we derive the following implications:

#### 2.3.1. Gas Composition

In this diagram, the gas flow (**G**) can be represented by a vector approach, so that the oxidation degree x corresponds to the direction of the vector $\vec{OG}$.

The directions of the main gaseous molecules encountered in direct reduction processes are shown in the diagram in Figure 3. In addition, pure carbon is also mentioned, although it is not a gas, in order to introduce the phenomena of carburization with depletion of carbon in the gas.

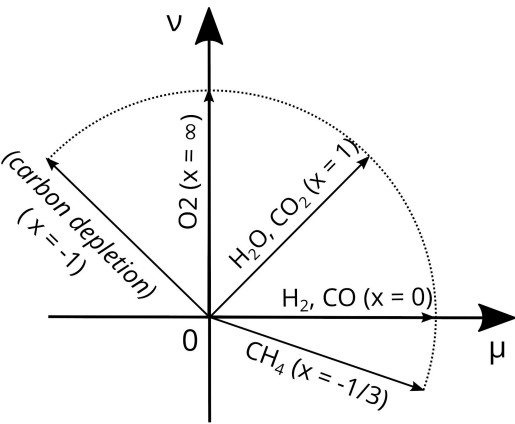

**Figure 3.** Directions of the main gases.

2.3.2. Gas Mixing

Both specific consumption and oxidation are extensive variables, so a simple law of additivity can be applied in the context of a gas mixture:

$$\mu_{mix} = \sum_{gas\,mixed} \mu_{gas} \tag{10}$$

$$\nu_{mix} = \sum_{gas\,mixed} \nu_{gas} \tag{11}$$

With a vector approach, the gas mixing process is simply described by Equation (12) and easily interpreted graphically, as shown in Figure 4.

$$\vec{OG}_{mix} = \sum_{gas\,mixed} \vec{OG}_{gas} \tag{12}$$

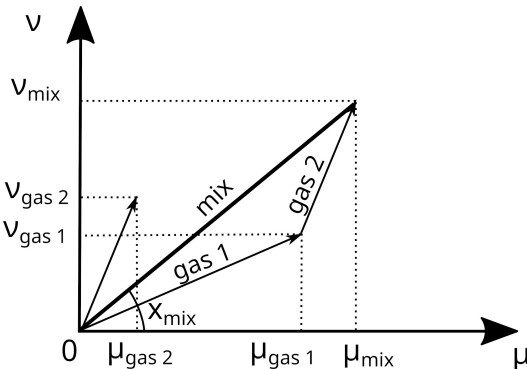

**Figure 4.** Graphical representation of the gas mixing.

2.3.3. Gas–Solid Reactions in the Shaft Furnace

Two main types of gas–solid reactions occur in direct reduction processes:

- Reduction: This reaction corresponds to the transfer of oxygen from the iron-bearing material to the reducing gas. In a counter-current gas–solid configuration, the oxygen variations in the ferrous burden and in the gas are strictly equal ($\delta\phi_O^s = \delta\phi_O^g$). Consequently, the variation in the gas-specific oxidation $\nu$ is equal to the variation in the burden oxidation degree y, defined by Rist [11] (1963) as the stoichiometric ratio O/Fe of iron oxides:

$$\delta\nu = \delta y \tag{13}$$

- Carburization: This reaction describes the transfer of carbon from the gas to the solid. The loss of carbon in the reducing gas is therefore related to the carburization rate of the DRI and can be calculated using Equation (14).

$$\delta\nu = -\delta\mu = \frac{w_C^{DRI}}{w_{Fe}^{DRI}} \cdot \frac{M_{Fe}}{M_C} \tag{14}$$

$w_C^{DRI}$ and $w_{Fe}^{DRI}$ are the mass fraction of carbon and iron in the DRI.

Finally, Figure 5 shows the graphical interpretation of these reactions in the mass balance diagram.

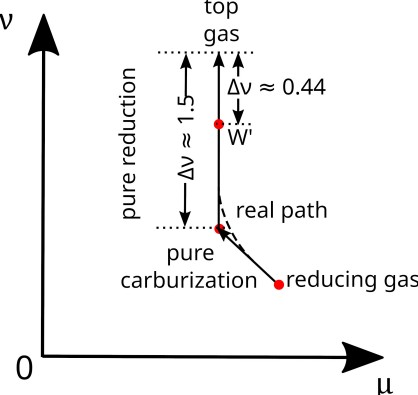

**Figure 5.** Graphical representation of reduction and carburization.

### 2.4. Thermodynamics Constraints

Similar to the development of the Rist diagram, it is possible to include thermodynamic constraints on the reduction mechanisms that were highlighted by Chaudron.

As a reminder, Chaudron established equilibrium conditions as a function of temperature between different iron oxides and the gas mixtures $H_2$–$H_2O$, CO–$CO_2$. The diagram thus represents gas–wustite and gas–magnetite thermodynamic equilibria as a function of the composition and temperature of the reducing gas, as shown in Figure 6.

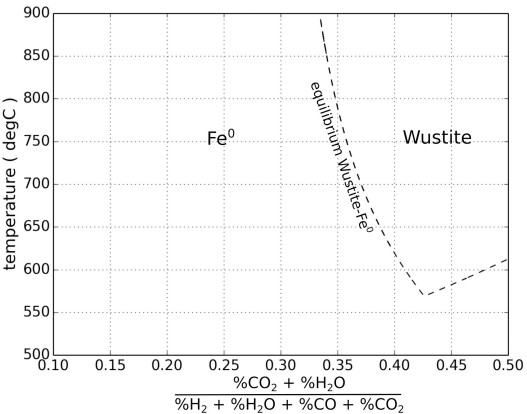

**Figure 6.** Graphical representation of the Chaudron (or Bauer–Glaessner) diagram for a typical reducing gas in the Midrex NG$^{TM}$ shaft.

We define the metallization line passing through the origin with a slope equal to $x_W$, which is the gas oxidation degree in equilibrium with Wustite and metallic iron. This value, $x_W$, is the oxidation degree of the so-called "Wustite Point" in the Rist diagram. Quatravaux

et al. [23] provided the methodology for calculating $x_W$ according to the Chaudron diagram:

$$x_{Chaudron} = \frac{\%CO_2 + \%H_2O}{\%CO + \%CO_2 + \%H_2 + \%H_2O} \tag{15}$$

$x_{Chaudron}$ is the oxidation degree of the reducing gas as defined in the Chaudron diagram formalism.

We then derive the corresponding oxidation degree for the mass balance diagram according to the approach presented by Quatravaux et al. [23], based on the following relationships:

$$x_W = (1 - \alpha) \cdot x_{Chaudron} - \alpha \tag{16}$$

$$\alpha = \frac{\sum \left(m + \frac{n}{2}\right) \%C_m H_n}{\eta_{CH}} \tag{17}$$

where:

$$\eta_{CH} = \%CO + \%CO_2 + \%H_2 + \%H_2O + \sum \left(m + \frac{n}{2}\right) \%C_m H_n \tag{18}$$

$\alpha$ is a coefficient to transpose the equilibrium lines from the Chaudron diagram to the corresponding gas oxidation degree defined by Rist, assuming the presence of hydrocarbons $C_m H_n$ in the reducing gas.

$\%C_m H_n$ is the $C_m H_n$ content of the gas in the reduction zone. It is assumed to be the same as the top gas.

The metallization line divides the reduction zone into two distinct parts for which the reduction conditions are thermodynamically different, as shown in Figure 7:

- If $slope > x_W$, metallization is thermodynamically impossible: this defines the pre-reduction zone
- If $slope < x_W$, metallization can occur: this defines the metallization zone.

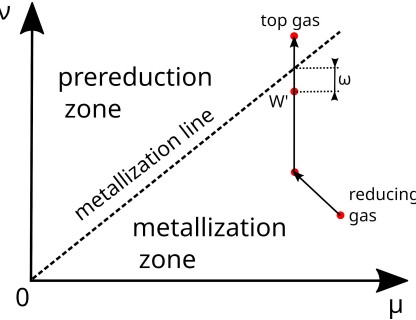

**Figure 7.** Graphical representation of the metallization zone and the deviation to ideality.

A detailed calculation of the metallization line has been presented by Quatravaux et al. [23]. The slope of the metallization line depends on the temperature and composition of the reducing gas. In the present study, we consider the local temperature in the schaft at the start of metallization. This is estimated to be 800 °C, in agreement with in-situ measurements by Takenaka and Kimura [27]. According to the definition of the Rist diagram, it is also possible to draw the deviation to ideality $\omega$, which is the difference between the iron oxidation degree at the beginning of metallization and the equivalent pure wustite point; the point is assumed to correspond to y = 1.056 for such a temperature, represented by the node $\mathbf{W'}$, as described in Figure 7.

$\omega$ must be positive to obtain a reduction in the iron oxides up to the metallization, so $\mathbf{W'}$ must be located in the metallization zone. Otherwise, the reducing power of the gas would be insufficient, and the reduction would stop at the wustite stage.

In conclusion, the metallization line—in particular, $\omega$—is an indicator that allows for visualization of the feasibility of a metallization according to the reducing gas injected in the

shaft furnace. It could be useful in the context of a prospective search for new breakthrough operating points.

## 3. Results

### 3.1. Reference Midrex NG$^{TM}$ Operating Point

For pedagogical purposes, the theoretical operating point of a Midrex unit as proposed by Sarkar et al. [28] (2018) serves as a reference. The extensive details provided in this reference outweigh the descriptions of operating points taken from various industrial plants, including those in Siderca (Argentina), Gilmore (United States) by Parisi and Laborde [24] (2004), and Khorasan (Iran) by Mirzajani et al. [29] (2018). We have completed the dataset under the following assumptions:

- The humidity of the fuel gas and process gas is 5 % and 10 %, respectively;
- The composition of natural gas is based on the work of Farhadi et al. [30] (2003);
- The natural gas injected directly into the vessel is assumed to be evenly distributed between the transition zone and the cooling zone;
- The injection of oxygen upstream of the bustle gas (called suroxygenation), although not specified by Sarkar, has been accounted for to maintain mass balance.

Table 1 lists the gas flow rates, while Table 2 lists their compositions. Data related to the DRI can be found in Table 3.

**Table 1.** Gas volume flow rates.

| Gas | Flow Rate (kNm$^3 \cdot$h$^{-1}$) |
|---|---|
| Fuel gas | 38.3 |
| Process gas | 78.5 |
| Process natural gas | 18.0 |
| Suroxygenation | 2.84 |
| Enrichment natural gas | 0.0 |
| Transition zone natural gas | 1.3 |
| Cooling zone natural gas | 1.3 |

**Table 2.** Gas composition.

| Composition | Bustle Gas | Dry Top Gas | Natural Gas |
|---|---|---|---|
| CO (%) | 35.0 | 21.4 | 0.0 |
| CO$_2$ (%) | 2.0 | 26.3 | 0.0 |
| H$_2$ (%) | 55.0 | 49.4 | 0.0 |
| H$_2$O (%) | 6.0 | 0.0 | 0.0 |
| CH$_4$ (%) | 1.0 | 1.7 | 87.6 |
| C$_2$H$_6$ (%) | 0.0 | 0.0 | 8.2 |
| C$_3$H$_8$ (%) | 0.0 | 0.0 | 3.1 |
| C$_4$H$_{10}$ (%) | 0.0 | 0.0 | 1.1 |
| N$_2$ (%) | 1.0 | 0.0 | 0.0 |
| $x$ | 0.069 | 0.242 | −0.342 |

**Table 3.** Production parameters.

| | |
|---|---|
| production | 100 t$\cdot$h$^{-1}$ |
| metallization rate | 92.7% |
| carburization rate | 2.2% |
| gangue | 5.8% |

### 3.2. Overview of the Mass Balance Diagram for a Midrex Unit

Figure 8 presents the corresponding graphical operating point described above. To facilitate the reading of the diagram, the reader is referred in parallel to the process description in Figure 1.

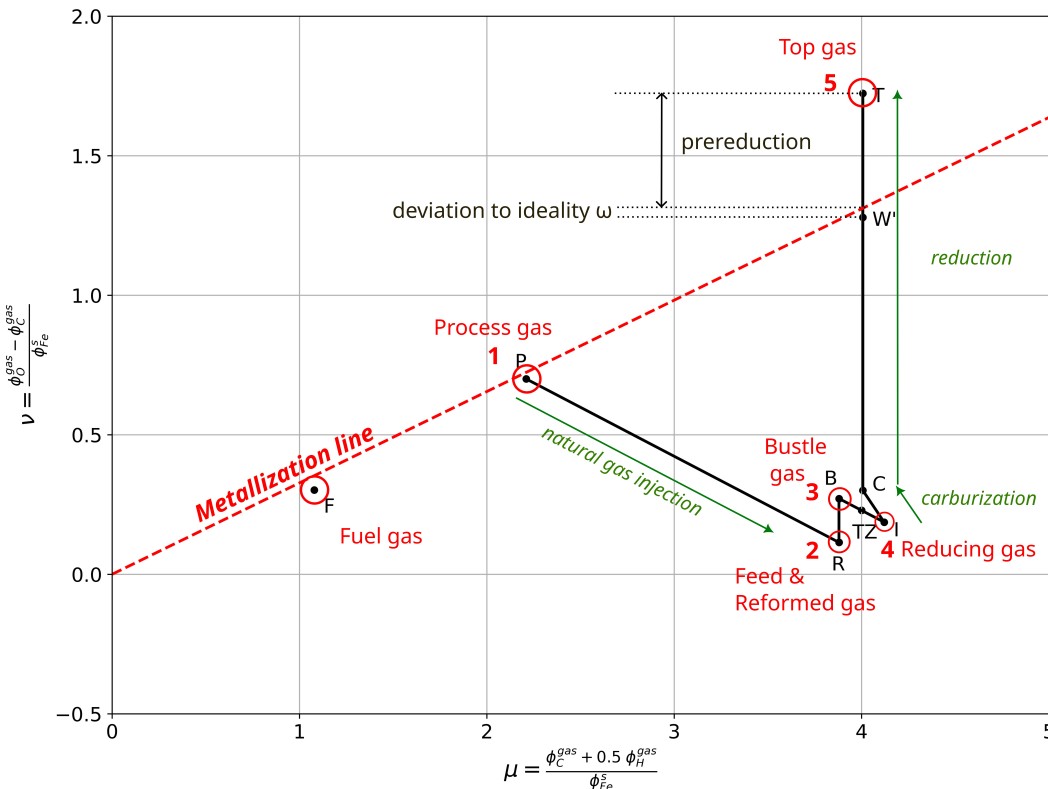

**Figure 8.** Balance diagram for a Midrex NG$^{TM}$ representative theoretical working point.

The operating point of the Midrex NG$^{TM}$ process is drawn through the following main steps:

1.  Process Gas (node 1):
    The flow and dry composition of the process gas are measured in the plant using the top gas analyzer, while the moisture content of the gas is estimated based on the vapor saturation pressure derived from the gas temperature. The first node corresponding to the process gas is positioned and labeled **P** in the diagram. The fuel gas, represented by the node **F**, is calculated and plotted using the same methodology.

2.  Feed Gas and Reformed Gas (node 2):
    The feed gas is a mixture of the process gas with some natural gas injections. The position of the feed gas is easily determined using the method described in Section 2.3.2 on gas mixing. The reformed gas node **R** overlaps the feed gas node because gaseous chemical reactions do not affect the node position.

3.  Bustle Gas (node 3):
    Bustle gas is a mixture of the reformed gas with natural gas and oxygen. The corresponding node **B** is drawn according to the mixing methodology developed in the Section 2.3.2. Note again that the internal reactions of the gas (combustion with oxygen and water gas shift equilibrium) does not affect the position of the node.

4.  Reducing Gas (node 4):
    The reducing gas, represented by node **I**, corresponds to the theoretical mixtures of all the gases injected into the furnace shaft below the gas–solid reaction zone. It includes the bustle gas, as well as some additional natural gas in the transition zone and in the cooling zone. Optionally, the additional steam accompanying the cooling gas

loop and the seal gas can also be considered. For practical reasons, we have added the node **TZ**, which corresponds to the theoretical mixture of the bustle gas with the natural gas injected in the transition zone.

5.  Top Gas (node 5):
    The methodology described in Figure 5 is used to define the graphical position of the top gas. First, the decarburized gas, shown as **C**, corresponds to the composition of the reducing gas after complete carburization and before reduction. We simplify the process description by assuming that carburization and reduction take place in separate zones. Carburization is achieved by in situ methane reforming and the Boudouard reaction. It is generally accepted that both reactions occur in the lower part of the metallization zone, in the transition zone, favored by the high local temperature and the presence of metallic iron acting as a catalyst as explained by Shams and Moazeni [25] (2015) and simulated by Hamadeh et al. [26] (2018).
    Finally, the top gas is deduced from the decarburized gas. Based on the reduction rate from pellets to DRI, we determine the position of the top gas node **T**, as well as the pure wustite point node **W′** defined in Section 2.4.

### 3.3. Application 1: Calculation of the Gas Moisture

A first application of this diagram is the quantification of gas humidity in Midrex NG^{TM} plants. This is possible if the gas composition is measured regularly, which is the case for the reformed gas, bustle gas, and top gas.

Let us first consider the wet gas as a mixture between its dry part and steam. The dry part of the gas is plotted on the diagram using the following method:

*   We start with the diagram describing the operating point of the industrial unit, as shown in Section 3.2 and illustrated in Figure 8. At this point, we recover the positions of all the wet gases on the diagram.
*   The gas analyzer gives the composition of the dry part of the gas. From this, we calculate the associated oxidation degree $x_{analyzer}$ and draw the "dry line" on the diagram, representing all the dry gases potentially associated with this composition:

$$\nu = x_{analyzer} \cdot \mu \tag{19}$$

*   In addition, we can perform a vector decomposition of the wet gas using the gas mixing methodology explained in Section 2.3.2. We thus define the "wet line" as the line with slope $x_{H_2O}$ equal to 1, passing through the node associated with the wet gas:

$$\nu = \nu_{wet} + x_{H_2O} \cdot (\mu - \mu_{wet}) \tag{20}$$

The node associated with the dry fraction of the gas is positioned at the intersection of the two lines defined above. The corresponding graphical construction is shown in Figure 9.

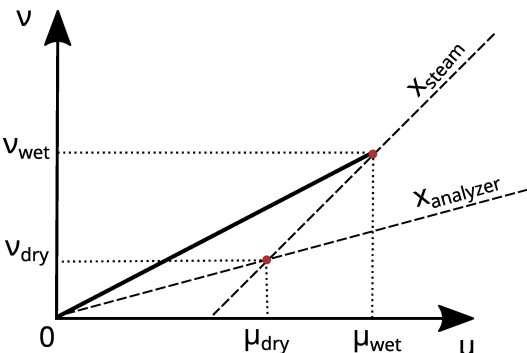

**Figure 9.** Graphical determination of the dry part of a gas.

Finally, the humidity of the gas can be calculated directly from the coordinates of the dry part of the gas according to the following relationship:

$$\%H_2O = 1 - \frac{1 - x_{wet}}{1 + x_{wet} \cdot (\mu_{dry} - 1) - \nu_{dry}} \tag{21}$$

Figure 10 shows the plot of the dry portions of the fuel gas, process gas, and top gas. These three nodes are located on the "dry top gas" line determined from the gas analyzer measurements. This shows the importance of the condensation phenomenon during the recycling of the top gas to the fuel and process gas.

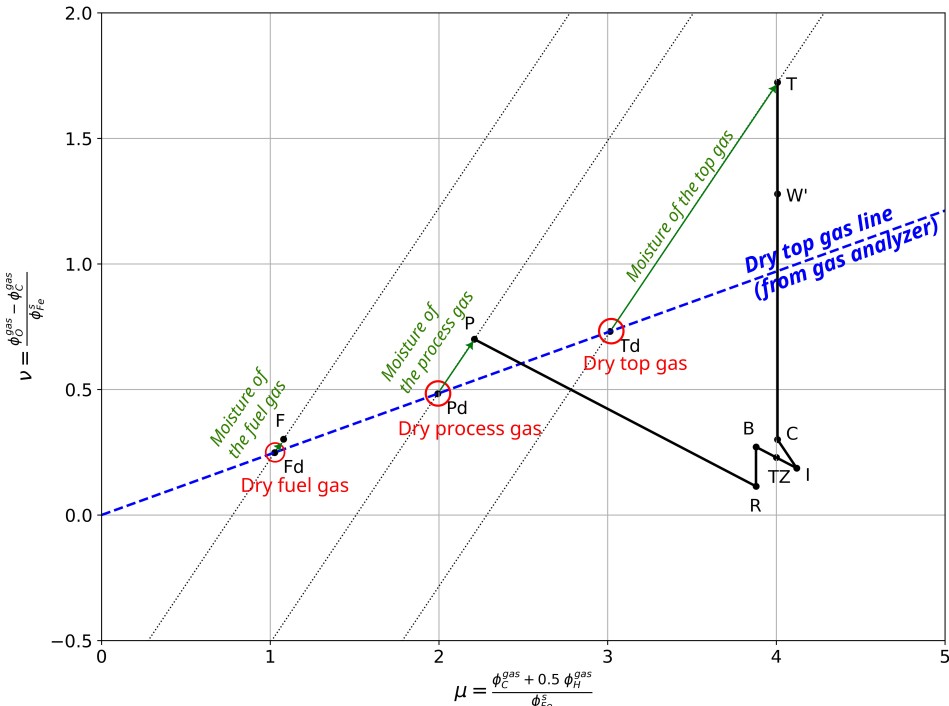

**Figure 10.** Graphical determination of the dry parts of the fuel gas, process gas, and top gas.

### 3.4. Application 2: Mass Balance of the Top Gas Recycling

A second application of this diagram concerns the mass balance of the top gas recycling. This approach can be used to check the consistency of the process operating point and thus diagnose the quality of the instrumentation measurements.

Since the amount of condensed water from the top gas is not measured, a wet gas mass balance is not possible. Therefore, we apply this approach to the dry fractions of the gas, as defined in the previous section.

We also assume that the flare gas is negligible in the mass balance. In other words, the top gas is completely recycled without any losses. The gas mass balance of the dry top gas is described by the following relationship:

$$Q_v^{dry\,top\,gas} = Q_v^{dry\,process\,gas} + Q_v^{dry\,fuel\,gas} \tag{22}$$

On the diagram, we identify the dry portions of the fuel gas, process gas, and top gas by their respective nodes designated by $\mathbf{F_d}$, $\mathbf{P_d}$, and $\mathbf{T_d}$ according to the graphical methodology shown in Figure 10. The mass balance corresponding to the dry top gas recycling is described by the following simple vector relationship:

$$\overrightarrow{OT_d} = \overrightarrow{OP_d} + \overrightarrow{OF_d} \tag{23}$$

This relationship is never strictly verified because of all the measurement uncertainties and the instrumentation drifts. Therefore, we calculate the equilibrium deviation on the top gas recycling mass balance (*DOTG*):

$$DOTG = 1 - \frac{\left\| \overrightarrow{OP_d} + \overrightarrow{OF_d} \right\|}{\left\| \overrightarrow{OT_d} \right\|} \tag{24}$$

*DOTG* is a simple and immediate quantitative criterion to diagnose the consistency of the main process measurements made on the plant, since all mass flow rates and the gas analyzer are involved in this calculation.

We illustrate the application of this criterion in the context of the ideal operating point described in Section 3.1. This is shown in Figure 11, where the relation (23) is perfectly validated. In addition, we use this criterion in the context of a 10% overestimation of the natural gas flow rate injected upstream of the reformer, as shown in Figure 12.

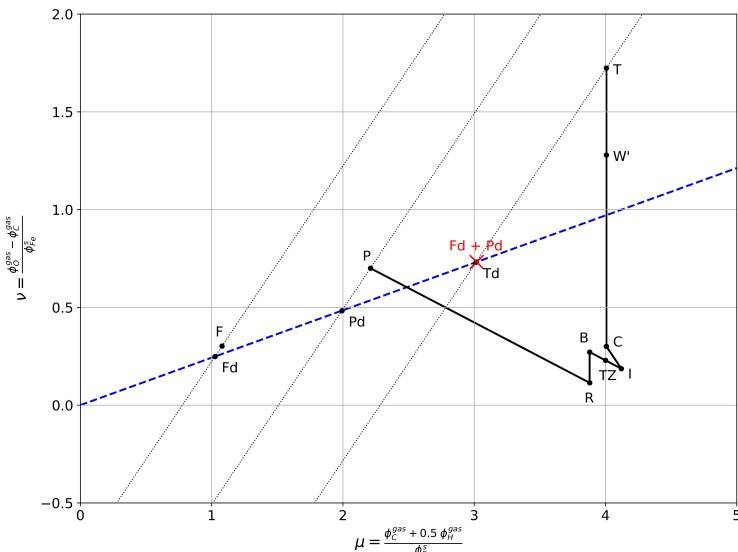

**Figure 11.** Ideal mass balance of the top gas recycling.

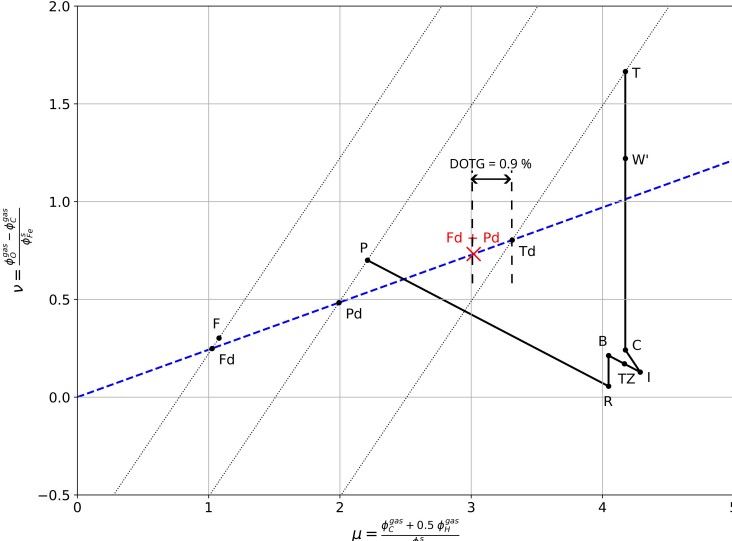

**Figure 12.** Influence of an overestimation of the injected natural gas flow measurement for feed gas preparation (+10%) on the deviation of the top gas recycling mass balance.

### 3.5. Application 3: Estimation of the Instantaneous Production Rate

The instantaneous hot metal production in the blast furnace is a parameter calculated from the difference in oxygen per unit of time between the inlet gas at the tuyeres and the outlet top gas at the throat. Since this difference is due to the reduction in the ferrous burden, it is then possible to deduce the associated hot metal production.

This parameter is a key piece of information for controlling the operation of the blast furnace, as it makes it possible to monitor the incoming flows (gas, coke, wind, ...) in terms of rate per ton of iron and provides an immediate information on any disturbance of the operating point.

The instantaneous production is therefore a direct consequence of the progress of the iron oxide reduction reaction in the process. This is different from hot metal production, which is measured by casting weight. These two concepts are relatively analogous in the context of operating balances. However, they diverge in the realm of process control, where data on the weight of castings are too inaccurate and too late.

In the case of the Midrex NG$^{TM}$ process, production is confused with the extraction rate of the DRI at the bottom of the shaft furnace. However, for similar reasons and problems as with the blast furnace, it is necessary to make a semantic distinction between (instantaneous) production and extraction.

As with the blast furnace, it is possible to estimate an instantaneous DRI production by calculating the oxygen balance on the reducing gas between the bottom and top of the reduction zone.

The mass balance diagram provides access to this information. This application is interesting for online use of the tool in order to detect fluctuations in the operating point and to make the necessary corrections to the process in a short time.

In order to apply this method, it is necessary to have previously established a material balance for the recycling of the top gas, whose methodology is described in the Section 3.4 and to quantify the DOTG drift.

In addition, we assume that the carburization and metallization rates are stable parameters over time with respect to variations in gas characteristics. This assumption is reasonable because the residence time of the material in the shaft furnace (several hours) is much longer than that of the gas (a few seconds). Therefore, we rely on the chemical measurements performed regularly on DRI.

We first determine the instantaneous reduction rate graphically based on the extraction rate. The corresponding methodology is shown in Figure 13.

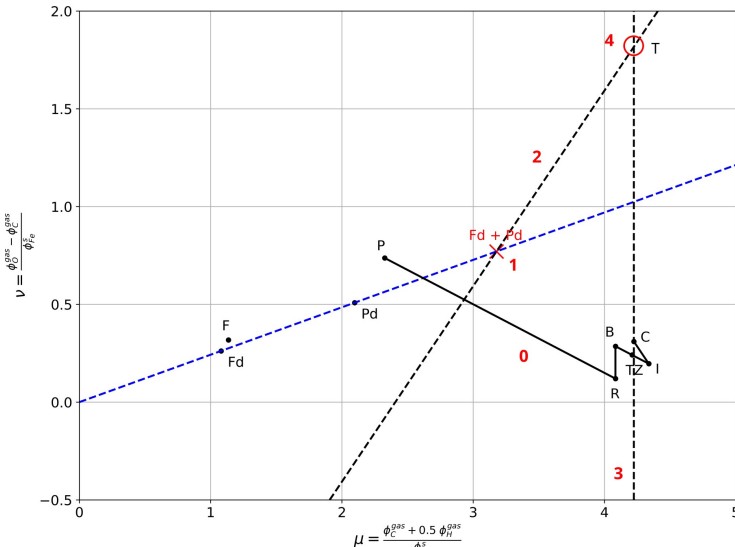

**Figure 13.** Determination of the instantaneous reduction rate.

The graphical methodology includes the following main steps:

0.　First, we draw the operating line of the Midrex NG$^{TM}$ working point from the process gas to the decarburized gas (nodes: **F**, **P**, **R**, **B**, **TZ**, **I** and **C**). At this point, the top gas is not shown on the diagram.

1.　We plot the node **T$_d$** associated with the dry top gas from the mass balance on the top gas recycling:

$$\overrightarrow{OT_d} = \frac{1}{1 - DOTG}(\overrightarrow{OF_d} + \overrightarrow{OP_d}) \tag{25}$$

2.　We draw the wet line of the top gas, passing through **T$_d$** with slope $x_{H_2O} = 1$.

3.　We also draw the line associated with the reduction path: it is a vertical line ($iso - \mu$) passing through **C**.

4.　We place the node associated with the top gas **T** at the intersection of the two lines drawn in steps 2 and 3.

We define the variation in the oxidation degree of iron obtained using this graphical construction based on the DRI extraction rate $\delta y_{extraction}$:

$$\delta y_{extraction} = \nu_T - \nu_C \tag{26}$$

The actual variation in the oxidation degree of the ferrous burden, based on instantaneous production, is derived from the following relationship:

$$\delta y_{production} = y_{pellet} - y_{DRI} \tag{27}$$

where $y_{pellet} \approx 1.5$ and:

$$y_{DRI} = 1.056 \times (1 - \text{metallization}) \tag{28}$$

The instantaneous production can be easily deduced from the assumed extraction and a correcting coefficient $\alpha$:

$$\text{production} = \text{extraction} \times \alpha \tag{29}$$

The graphical construction described in Figure 13, plotted according to the instantaneous production, would allow the Equations (26) and (27) to be equal. The correction coefficient is calculated with the following equation:

$$\alpha = \frac{\delta y_{extraction} + 2(\nu_C - \nu_I)}{\delta y_{production} + 2(\nu_C - \nu_I)} \tag{30}$$

It should be noted that the vector $\overrightarrow{IC}$ corresponds to the rate of carburization of the DRI. Therefore, its length is independent of the iron molar flux chosen for the graphical representation.

## 4. Conclusions

In this paper, we have presented a new graphical tool developed for the direct reduction shaft processes, inspired by the Rist diagram originally dedicated to the blast furnace.

Any gas flow in the plant can be represented on this diagram by a vector whose components, called specific consumption and specific oxidation, represent certain stoichiometric gas/iron ratios.

This particular graphical tool has several unique features. First, the chemical composition of each individual gas is responsible for determining the direction of the associated vector. In addition, a simple vector addition can accurately represent successive gas mixtures. Finally, this tool allows direct graphical representation of the reduction and carburization rates associated with direct reduced iron (DRI).

In addition, the thermodynamic conditions for reduction, highlighted in the Chaudron diagram, are plotted in this tool and are used in a manner strictly equivalent to the Rist diagram.

Several applications of this tool are presented:

- When the composition of the (dry) gas is measured, it is possible to quantify its moisture.
- A graphical mass balance of the top gas recycling allows diagnosis of the consistency of the measurements of flowmeters and gas analyzers.
- The evaluation of the instantaneous production of DRI, as of the instantaneous production of hot metal for the blast furnace, is obtained from a graphical mass balance on the reducing gas at the inlet and outlet of the shaft.

As a result, this graphical tool can be useful for production teams to monitor and optimize the process flow. In this sense, it would benefit from being used online.

This tool therefore promotes dialog within a community of students, as well as engineers, technicians, and operators, in order to better understand the process and its optimization.

In this context, it would be of great academic and industrial interest.

**Funding:** This research received no external funding.

**Conflicts of Interest:** The author declares no conflict of interest.

### Abbreviations

The following abbreviations are used in this manuscript:

Greek letters

| | |
|---|---|
| $\mu$ | specific consumption |
| $\nu$ | specific oxidation |
| $\phi$ | molar flux (mol·m$^{-3}$·s$^{-1}$) |

Latin letters

| | |
|---|---|
| $a_j$ | volumetric fraction of the molecule $j$ in a mix gas |
| $c_i$ | local molar concentration of the atomic element $i$ (mol·m$^{-3}$) |
| $M_i$ | molar mass of the element $i$ (g·mol$^{-1}$) |
| $n_i^j$ | number of atoms $i$ in the molecule $j$ |
| $Q_m$ | mass flow rate (kg·s$^{-1}$) |
| $Q_v$ | volumetric flow rate (m$^3$·s$^{-1}$) |
| $V_m$ | molar volume of the gas (m$^3$·mol$^{-1}$) |
| $w$ | mass fraction |
| $x^{gas}$ | gas oxidation degree (derived definition) |
| $X^{gas}$ | gas oxidation degree (original definition) |
| $y$ | burden oxidation degree |

superscript

| | |
|---|---|
| gas | related to gaseous element |
| s | related to solid element (iron bearing material) |

subscript

| | |
|---|---|
| $i$ | atomic element (C, H, O or Fe) |

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
