# Peer review of "A Graphical Tool to Describe the Operating Point of Direct Reduction Shaft Processes"

_metals, doi:10.3390/met13091568_

Round 1

Reviewer 1 Report

Comments and Suggestions for Authors

The authors presents a very well written manuscript on a highly relevant topic. The manuscript can be accepted for publication as is, with just one minor issue, that could be addressed in the publication process:

Section 1 Introduction

Please check second sentence "DRI accounts for 10% of total DRI production"? Should this be "of total iron production"?

Author Response

Dear Reviewer,

Thank you for reviewing this article. 
The original sentence was wrong, I have corrected the article according to your comment.

Sincerely,

Reviewer 2 Report

Comments and Suggestions for Authors

The graphical tool presented in this paper is an innovative approach to analyse a direct reduction shaft process. It is very useful to monitor and analyse the operation shaft furnace. The content is worth to be published.

I have some small comments which might be considered in the final revision.

The Chaudron diagram is also known as the Baur-Glaessner diagram. It is suggested to mention this once in the paper.

Equation (8)

The definition of the oxidation degree of RIST, which I know, is (O+H2)/(C+H2). It is the abscissa of the RIST diagram for the system Fe-C-H-O. In which paper did RIST publish equation (8)?

Figure 8

Node F: Fuel gas is a mixture of top gas and natural gas. How can this be identified in the diagram?

Node TZ and I: What is the difference between these two nodes? Is it the moisture of the cooling gas?

Equations (16) to (18)

Describe the meaning of the variable α.

Author Response

Dear Reviewer,   Thank you for reviewing this article.  Several corrections have been made in this new version to address the suggestions for improvement. I have also provided some answers and comments to your questions in the text below.   Sincerely,   ------------------- Comment 1 : The Chaudron diagram is also known as the Baur-Glaessner diagram. It is suggested to mention this once in the paper.   Response 1: I have added this information to the legend of Figure 6.   ------------------- Comment 2: Equation (8) : The definition of the oxidation degree of RIST, which I know, is (O+H2)/(C+H2). It is the abscissa of the RIST diagram for the system Fe-C-H-O. In which paper did RIST publish equation (8)?   Response 2: The definition given in (8) was not explicitly given by Rist. However, this value, denoted by x, corresponds to ((O+H2)/(C+H2) -1) : x = X -1 where X = (O+H2)/(C+H2) The relation (8) was deduced from the work of Rist, according to the relation between x = ((CO2 + H2O) /(CO+ CO2 + H2 + H2O)) and X, the definition you mentionned. x was given in most of his articles (cf : references 11 and 12 for example). In a previous article (ref 23 : https://www.mdpi.com/2075-4701/11/12/1953, section 2.3), I explained this relationship (8) as a starting point for adapting the Rist diagram to DRI conditions.    ------------------- Comment 3 : Figure 8 :  Node F: Fuel gas is a mixture of top gas and natural gas. How can this be identified in the diagram? Node TZ and I: What is the difference between these two nodes? Is it the humidity of the cooling gas?   Response 3: Node F:  The fuel gas is the part of the top gas that is recycled to heat the reformer and preheat the reducing gas. The "Fuel Natural Gas" is injected downstream as shown in Figure 1.    Node TZ and I: As you may know, some natural gas is injected in two zones in the shaft: 1) in the transition zone, it is called transition zone natural gas. It contributes directly to the gas-solid reactions (carburization + reduction). 2) at the bottom of the cooling zone, to feed the cooling circuit and to compensate the part of the cooling gas that flows to the upper part instead of in the recirculating cooling circuit.   To visualize the balance of injected natural gas in each zone, I defined and plotted TZ:  1) The length BTZ corresponds to the flow rate of the natural gas transition zone.  2) The length TZI corresponds to the flow rate of the natural gas cooling zone.      ------------------- Comment 4 : Equations (16) to (18)   Describe the meaning of the variable α.   I have added additional information about the meaning of α and I have also added the reference of the previous article dealing with this coefficient (reference 23 : https://www.mdpi.com/2075-4701/11/12/1953, section 2.5).

Reviewer 3 Report

Comments and Suggestions for Authors

Author Response

Dear Reviewer,

I would like to thank you sincerely for your detailed proofreading of this article, which has improved the quality of the document.
I have also provided some answers and comments to your questions in the text below.

Sincerely,

-------------------
Comment 1 : Properly describing them can be tiresome. Please check the equations AND Abbreviations for consistence. 
As an example see equation (2) AND units for molar flow rate (mol/s). Equation (1) should be dimensionless: such as defined it is not.

Response 1 : I have checked the units of the equations you mentioned, I am afraid I did not find any error, the units seem to be correct.
I also confirm that equation (1) is dimensionless.

-------------------
Comment 2 : The symbol for “specific consumption” is missing on figure 3.

Response 2 : figure 3 is now corrected.

-------------------
Comment 3 : If carburation is not happening the amount of Carbon in the gas should be constant: %CO + %CO2 + m %CmHn. 
Carburation means some of this carbon is solubilized, there is a variation on the sum above mentioned. 
How does equation (14) describe that? Equation (14) only applies if the amount of CmHn is negligible.

Response 3 : 
The amount of carbon depletion in the gas, mol_C / mol_Fe, is calculated directly from the chemical composition of the DRI. 
This information is useful to estimate the amount of carbon in the gas in the reduction zone after carburization phenomena.
The corresponding value is calculated using relation (14). 
Equation (14) gives the length of the segment [IC] in Figure 8. If there is no carburization, nodes I and C are superimposed.

I explained this part in a previous article (reference 23: https://www.mdpi.com/2075-4701/11/12/1953), section 2.4, where I detailed a simplified case without carburization and the reality where carburization occurs. 

Furthermore, carburization occurs with or without hydrocarbons, due to the Boudouard reaction (2CO -> CO2 + C) or in-situ reforming.
Carbon in DRI is present as graphite deposition or through cementite.

-------------------
Comment 4 : Wustite point which is given as O/Fe = 1.056 (line 159) is temperature dependent (the non stoichiometry is variable): the temperature should be given.
It may be cumbersome to rewrite but the X axis of figure 2 should read and not since has not been defined.

Response 4 : 
You are right, I have added in the article that I assume O/Fe = 1.056 at 800 °C, the estimated temperature at which metallization starts, according to probing measurements.
All this information was added in lines 168 to 174. 

-------------------
Comment 5 : The Y axis is also mistyped (mistyping means the author does not keep the same symbology along the text). 
Check all figures for mistakes on X or Y axes symbology: X axis at figures 8, 10 , 11, 12 and 13 is mistyped. Uniformity for sake of clarity.

Response 5 : Thank you for this remark. I have updated almost all figures according to you comment.

-------------------
Comment 6 : Hot metal instead of pig iron, line268 and 271and 276 and 337.

Response 6 : Corrections made.